

# A Climatological Perspective on Cyclones and Surface Impacts in the Eastern Mediterranean Using Potential Vorticity-Based Classification

Tali Sarit Gens[1], Leehi Magaritz-Ronen[1], and Shira Raveh-Rubin[1]

[1]Department of Earth and Planetary Sciences, Weizmann Institute of Science, Rehovot, Israel

**Correspondence:** Tali Sarit Gens (tali-sarit.gens@weizmann.ac.il)

**Abstract.** Eastern Mediterranean Cyclones (EMCs) are a major contributor to extreme weather in their region, including precipitation, strong winds, cold extremes or dust storms, significantly impacting the population and natural environment. Understanding the relationship between EMC variability and associated impacts is key to understanding their predictability and forecasting them accurately. Various processes come together to govern the genesis and development of EMC, affecting

eventual cyclone characteristics and impacts. These processes have distinct signatures on the potential vorticity (PV) distribution. Existing approaches for EMC classification provide limited physical interpretations of cyclone variability, associated impacts and predictability. Here we classify EMCs based on their associated upper-tropospheric PV structures providing a novel process-based framework for EMC classification. Using Self-Organising Maps classification of ERA5 reanalysis data, we find 6 coherent PV patterns that typify cyclone clusters, each with its own signature precipitation pattern, which we quantify

using ERA5 forecasts, IMERG and local station data. For each of the seasons, there are dominant clusters that lead to extreme precipitation and temperatures. In particular, two clusters with high PV values dominate the eastern Mediterranean's annual precipitation. Evidently, a strong ridge upstream of the PV trough has a greater impact on precipitation than the PV pattern with a weak ridge upstream. Temperature anomalies and extremes during cyclone passage were found to be strongly linked to upper-level PV patterns, with certain cyclones types causing significant near-surface hot or cold extremes (or both). While

we found that the overall frequency of cyclones shows no significant trend, specific cyclone types display notable trends, with some increasing and others decreasing, which may suggest a rise in the frequency of drier cyclones if these trends continue, potentially indicating a shift toward drier conditions that could impact precipitation patterns. This classification approach enhances our understanding of the link between cyclones variability and their surface impacts in the region through processes reflected in upper-level PV distributions. These findings provide a framework for systematic evaluation of cyclones and their

prediction, and could benefit strategies for managing the societal and environmental impacts of EMCs at weather and climate timescales.

## 1   Introduction

The Mediterranean is known as a climate change hotspot (Tuel and Eltahir, 2020; Giorgi and Lionello, 2008), with the Eastern Mediterranean region experiencing persistent hot and dry summers, and more variable temperatures and rainfall events in



winter (Kushnir et al., 2017). The region is influenced by mid-latitude, subtropical, and tropical weather systems that can bring extreme weather to the region (Alpert et al., 2005), among them, notorious floods and cold spells (Dayan et al., 2021; Hochman et al., 2022b). Such storms bear tremendous damage to infrastructure and human lives in the region (Hochman et al., 2022a). While general drying is expected in the future, there is yet high uncertainty with regard to changes in extreme precipitation in the region (IPCC, 2021).

Eastern Mediterranean Cyclones (EMCs) are the major contributor to extreme weather in this region, including precipitation (Sharon and Kutiel, 1986; Alpert et al., 1990; Ziv et al., 2006), strong winds (Raveh-Rubin and Wernli, 2015, 2016; Nissen et al., 2010), cold extremes (Klaider and Raveh-Rubin, 2023), or dust storms (Nissenbaum et al., 2023) and compound events (Portal et al., 2024). As such, there is tremendous value in accurate and timely predictions of EMCs in this densely populated area. From a climatological perspective, it has been shown that the frequency of EMCs is decreasing (Maheras et al., 2000;

Zappa et al., 2015), along with a drying trend in mean precipitation over the region. However, the response of extreme rainfall events remains uncertain, with some studies suggesting an increase in intensity or frequency (Drori et al., 2021; Philandras et al., 2011; Alpert et al., 2002; IPCC, 2021). This apparent contradiction, often referred to as the 'Mediterranean precipitation paradox,' has been discussed in previous studies, highlighting the complex interactions between large-scale circulation changes and regional precipitation patterns (Alpert et al., 2002; Alpert, 2004; Zappa et al., 2015).

To improve our understanding of present and future extreme weather events in the Eastern Mediterranean, it is therefore crucial to understand the drivers shaping EMCs and their associated spectrum of hazards, including extremes. The Mediterranean Basin is known as a cyclogenesis region (Campins et al., 2000), with multiple processes involved in the genesis and evolution of Mediterranean cyclones throughout the basin. Most cyclones form through baroclinic instability, induced by disturbances to the upper-tropospheric westerly flow (Stevens et al., 2013; Egger, 1995). Additionally, diabatic processes, air-sea heat exchange,

and topographic forcing play crucial roles in cyclone formation and deepening (Avolio et al., 2024; Stathopoulos et al., 2020; Shay-El and Alpert, 1991; Mattocks and Bleck, 1986) as well as in the formation of their associated precipitation (Flaounas et al., 2022).

In the Eastern Mediterranean, various types of cyclones exist, each characterised by distinct formation and evolution processes. One prominent winter cyclone is the Cyprus low, which develops in the cyclogenesis hotspot near Cyprus, south of

the Turkish Plateau and is associated with baroclinic instability (El-Fandy, 1946; Alpert and Shay-El, 1994; Goldreich, 2003). "Sharav" lows typically develop in spring due to surface thermal anomalies over land, usually forming in the lee of the Atlas Mountains before rapidly propagating over North Africa and the Mediterranean (Trigo et al., 1999; Alpert and Ziv, 1989; Saaroni et al., 1998; Goldreich, 2003). The Red Sea Trough, another key system, can evolve into a closed pressure minimum under favourable atmospheric conditions, such as an upper-level trough; significantly contributing to regional precipitation and

occasionally triggering dust storms (Goldreich, 2003; Tsvieli and Zangvil, 2007; de Vries et al., 2013; Awad and Mashat, 2019; Ziv et al., 2022). Several case studies have examined the dynamics of these systems, together highlighting the variability in the development mechanisms and impact on surface weather (Raveh-Rubin and Wernli, 2016; Buzzi and Tibaldi, 1978; Flocas, 2000; Alpert and Ziv, 1989; Alpert et al., 1999; Krichak et al., 1997). Given the differences in cyclone formation processes, it is expected that their associated impacts, such as precipitation, temperature anomalies, or dust transport, also vary system-



atically with cyclone type as has been shown for the Mediterranean region as a whole (Givon et al., 2024). To gain a broader understanding of cyclone variability beyond individual case studies, Eastern Mediterranean cyclones have been classified based on their location, intensity, and morphology (Flocas et al., 2010; Almazroui et al., 2015; Saaroni et al., 2010; Maheras et al., 2000) mostly focusing on their surface-related features. These studies have shown that cyclones forming in different regions and experiencing different pathways exhibit distinct seasonal and interannual variability, with Cyprus Lows dominating winter

precipitation, while Sharav lows and Red Sea Troughs playing a larger role in transitional seasons. Additionally, some studies have assessed cyclone predictability, revealing that forecast skill varies depending on the synoptic environment and cyclone type (Maslova et al., 2021).

  While several studies have explored cyclone impacts, most have considered cyclones as a general atmospheric feature rather than differentiating between cyclone types. As a result, it is not clear how specific cyclone types influence precipitation distri-

bution, temperature extremes, and dust transport in the region. More specifically, there is a lack of generalised understanding of the upper-level forcing of EMCs (Zangvil et al., 2003; Ziv et al., 2006). Here, we aim to address the gap by classifying EMCs based on their upper-level PV distribution. This classification will allow us to gain valuable insights into the development mechanism and impacts of each one of the groups of cyclones in the region. Such process-based classification is essential to gain a comprehensive understanding of the natural variability of the regional cyclone formation and characteristics, shaped by

the upper-level mechanisms. Specifically, we address the following questions:

1. What are the recurring patterns of upper-level PV that govern cyclone formation in the Eastern Mediterranean? Can these patterns describe the regional variability?

2. Are the recurring PV patterns related to the observed seasonal variations of different types of EMCs?

3. Are upper-level PV patterns linked to distinct surface impacts such as precipitation and extreme temperatures during
EMCs?

4. Are there observed trends for PV patterns during EMCs?

  To address these questions, we analyse 686 EMC tracks using ECMWF reanalysis ERA5 from 1979 to 2020. Upper-level PV in the eastern Mediterranean is used to classify the EMCs. Section 2 details the methodology employed in this study, including the Self-Organising Map (SOM) classification technique and the data sources used for classifying EMCs. Section 3 presents

the six identified cyclone types, their upper-level PV patterns, associated surface impacts, and observed trends. Finally, section 4 discusses the broader significance of these findings, offers recommendations for future research, and concludes.

## 2 Data and Methods

To understand the variability of EMCs, we classify cyclone tracks by the upper-level PV pattern at the time of their minimum sea-level pressure (SLP) employing the self-organising map (SOM) algorithm, and analyse the respective precipitation and

2-m temperature fields. In the following, we provide details on the data and methods used.



## 2.1 Atmospheric data

The fifth reanalysis generated by the European Centre for Medium-Range Weather Forecasts (ECMWF), ERA5 (Hersbach et al., 2020), uses the Integrated Forecasting System (IFS, cycle 41r2). In this study, we used ERA5 data from 1979 to 2020 in the Eastern Mediterranean domain (27°N–53°N, 7°E–38°E), interpolated to a horizontal resolution of 0.5°. Specifically, we analysed PV on isentropic levels between 320 K and 340 K at 5-K intervals, SLP, precipitation of both convective and large-scale components, and 2-meter temperature.

Cold and hot extremes were identified based on the 2-meter temperature, averaged daily to eliminate diurnal differences as a factor in temperature variability. To identify the strongest daily deviations across all seasons, we considered at each grid point the daily 2-meter temperature anomalies (deviations) from the long-term (1979-2019) monthly mean. A cold or hot daily extreme was identified as a day that falls within the 5th or 95th percentile of the strongest anomalies at each grid cell, respectively (Klaider and Raveh-Rubin, 2023).

## 2.2 Precipitation data

We quantify precipitation using a combination of three data sources.

– Short-term forecasts of precipitation are taken from ERA5, providing 1-h accumulated large-scale and convective precipitation from 12 to 18-h forecasts. Shorter lead times are avoided due to potential spinup biases.

– Satellite-based precipitation observations are provided by the Integrated Multi-satellitE Retrievals for GPM (IMERG) (Huffman et al., 2015), which combine data from core satellites (such as the GPM Core Observatory), a microwave constellation, infrared sensors, and surface precipitation gauges from 2000 to 2020. IMERG provides high spatial (0.25°) and temporal (half-hourly) resolution precipitation estimates, improving accuracy and filling gaps in remote regions.

– Daily accumulation of precipitation from 6 am to 6 am in 60 stations in Israel is available from the Israel Meteorological Service (IMS) for 1979-2017. This homogeneous data set (Yosef et al., 2019) has been adjusted to account for biases and inconsistencies in observations over time.

ERA5 precipitation data were validated against IMERG satellite-based observations to ensure their reliability for climatological analyses conducted in this study. This validation allows us to confidently extend our analysis to a longer climatological period (1979–2020) and to analyse a larger sample of cyclones. Additionally, the ERA5 dataset enables differentiation between convective and large-scale precipitation, provided in ERA5 data, facilitating a detailed examination of precipitation mechanisms associated with each cyclone cluster. We analyse ERA5 precipitation accumulated for 3 days, beginning at 6 am of the day of minimum cyclone SLP to match the daily accumulated station data.

## 2.3 Combined Cyclone Tracking Algorithm

Considering the variability that exists among cyclone identification and tracking methods, which lead to diverging results in the Mediterranean, here we used a robust cyclone track dataset that takes into account several methods. These composite tracks



combine outputs from ten cyclone-tracking approaches (Flaounas et al., 2023) from the period 1979-2020, derived from the ERA5 reanalysis hourly intervals (Hersbach et al., 2020). We select composite tracks with a confidence level of 5, indicating that at least 5 cyclone detection methods agree on the cyclone track. This choice is a compromise between too few resulting tracks for a higher confidence level and over-representation of spurious summer cyclones for lower confidence intervals. We then select cyclone tracks that are found within the domain of 27-38°N, 27-38°E (Figure 1, panel b) for at least one time step, resulting in 686 cyclone tracks. The peak time of the cyclone is defined based on the minimum SLP along the track.

### 2.4 Self-Organising Map (SOM) classification

We classify cyclones during their peak intensity (minimum SLP) by analysing upper-level isentropic PV fields. We focus on the 320-340K isentropic surfaces at 5-K intervals, vertically averaged within the fixed extended geographical region of 27°N to 53°N and 7°E to 38°E. This domain is more extended than the domain for the surface cyclone features, to include the extent of all upper-tropospheric large-scale PV features that can affect the cyclone domain. Using a self-organising map (SOM) algorithm, provided by MathWorks, we classify the PV patterns. Each EMC member is eventually attributed to one of the clusters, based on their PV characteristics. Six clusters were found to represent distinct recurring PV patterns. These six clusters effectively represent the distinct upper-level atmospheric states. Increasing the number of clusters does not reveal new patterns; instead, it dilutes the significance of our impact analysis. Conversely, selecting a lower number of clusters may enhance the statistical significance of the results, but it fails to capture the full range of upper-level atmospheric states present in this region during cyclone events.

Since the upper-level PV structure associated with cyclones exhibit substantial variability, some cyclones may be assigned to different clusters in different runs of the SOM. This variability can cause individual PV patterns to shift between two clusters across different algorithm iterations. To mitigate this issue and ensure a more robust classification, we ran the SOM algorithm 100 times. The cluster assignment for each cyclone was determined based on the most frequently occurring classification across these runs, requiring that the selected cluster appeared in at least 60 runs. This statistical approach reduces sensitivity to minor variations in clustering and provides a more stable and representative classification of the upper-level PV structures. Notably, only 11 cyclones changed their cluster assignments between runs, indicating a high degree of stability in our final classification.

### 2.5 Cyclone deepening rate

We diagnose the maximum deepening rate of EMCs using units of Bergeron (Sanders and Gyakum, 1980). This rate is normalised by the cyclone latitude and is given at time $t$ by:

$$\text{Bergeron}_t = \frac{\sin(60°)}{\sin(\phi_t)} \left( \frac{\text{SLP}_{t-12h} - \text{SLP}_t}{12h} \right)$$

where $\phi_t$ is the latitude of the cyclone center at time $t$ , $\text{SLP}_{t-12h}$ is the sea level pressure (SLP) 12 hours before time $t$ and $\text{SLP}_t$ is the sea level pressure (SLP) at the time $t$.





Different from a commonly used 24-h window for cyclones over oceans, we employ a 12-hour window due to the relatively short tracks. The window is centred around time $t$ and moved along each cyclone track to account for the maximal deepening rate for each track. Cyclones with deepening rates greater than 1 Bergeron are classified as explosive cyclones.

## 2.6 Significance of the trends

To evaluate long-term trends of EMC frequency while reducing short-term inter-annual variability, we first smooth the data using a 3-year moving mean. The statistical significance of long-term trends is evaluated with the Mann-Kendall (MK) test for the smoothed data, using a significance level of the test, $\alpha = 0.1$ and resulting in a binary result of 1 indicating a significant trend, or 0 if not.

It is important to note that the trends are calculated based on a relatively small number of cyclones (order of 100), which accurately represent the actual occurrence of this cyclone type in our region. However, despite the small sample size, the trend may indicate the beginning of a pattern which may continue in the warmer future. Therefore, when interpreting the trends, this limitation should be considered, particularly regarding long-term changes and statistical significance.

The analysis is based on rainy seasons (1 August–31 July) rather than calendar years. Therefore, the first and last seasons in the record (e.g., the 1979 and 2020 seasons) are partially truncated, and the trend calculation does not include the calendar year 2020, as the corresponding rainy season extends beyond the dataset period.

## 3 Results

### 3.1 Cyclone characteristics

A total of 686 cyclones were identified entering the study domain during the analysis period from 1979 to 2020. Figure 1 summarises the key climatological characteristics of these systems. Panel (a) shows the annual frequency of cyclones, revealing substantial inter-annual variability but no significant long-term trend according to the MK test. The climatological mean track density (Fig. 1b) highlights preferred regions of cyclone presence, particularly north of Cyprus and along the Turkish coast. Cyclone occurrence is enhanced during the winter and early spring months, with peak activity from December to April at 14% occurrence frequency (Fig. 1c). These EMC characteristics are in line with known regional patterns (Nissen et al., 2010; Maslova et al., 2021; Alpert et al., 2004; Flocas et al., 2010), and substantiate the applicability of our chosen EMC track data.





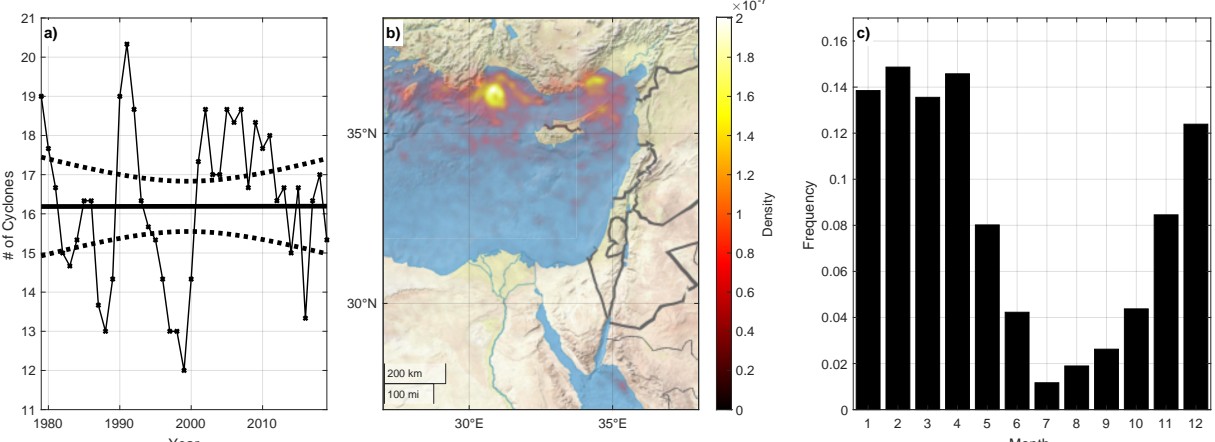

**Figure 1.** (a) Time series of the smoothed 3-year moving mean of cyclone seasonal occurrence from 1979 to 2020, markers indicate the total number of cyclones per year (for the precipitation season 1 August–31 July). The thin solid line connects the smoothed yearly values, and the thick solid line represents a linear trend fitted using the fitlm function in MATLAB. Dashed lines indicate the 95% confidence interval of the linear fit. (b) Geographical climatological cyclones track density of all 686 cyclones (shading). (c) Monthly distribution of cyclone occurrences. The relative frequency is normalised relative to the number of cyclones, i.e., sums up to one.

## 3.2 SOM classification

Six meaningful clusters are revealed from the SOM analysis, organised in a so-called map that reflects gradual transitions between cyclone types, with neighbouring clusters sharing similar PV distributions. Most clusters are characterised by high PV in the upper troposphere, typically organised in a trough-like structure upstream of the cyclone centre. Each cluster is
180 characterised by the composite mean PV and SLP of its cyclone members (Figure 2). Cluster 1 is marked by high PV values and strongly-tilted wave-breaking structure, while the neighbouring Cluster 3 exhibits a similar PV pattern but with weaker wave-breaking and PV values and less pronounced ridge upstream, over the central Mediterranean and Europe. In contrast, Cluster 2, which includes the largest fraction of members (22% of the cyclones), has local PV values comparable to values seen in Cluster 1 over the Middle East but is marked by a wide and deep trough and no wave breaking. Cluster 4 shows a more
185 zonal PV distribution with lower PV and PV gradients compared to its neighbouring cluster 2. Clusters 5 and 6, which are the least populated (13% and 10% of the cyclones, respectively), have relatively low PV, although Cluster 5 is still associated with wave breaking.

Considering the composite SLP distribution, Cluster 1 shows a local minimum in SLP over Cyprus, indicative of relatively deep cyclones, with a composite mean central pressure of 1008.6 hPa. Cluster 3 is similar but has, on average, a shallower
190 (1009.7 hPa) and more northeastern cyclone centre location, with a markedly less pronounced ridge upstream. Cluster 2 has the deepest SLP minima (1007.2 hPa), located west of Cyprus, while Cluster 4, with a shallower intensity (1008 hPa), still exhibits a distinct low in the climatological cyclogenesis region near Cyprus. Differently, Clusters 5 and 6 show SLP minima in the southeast Mediterranean, although without a clear closed low within the domain, suggesting more variability in the





location of the cyclone centres. Indeed, fig. 2 shows that the location of the minimum SLP of individual tracks has a larger
195  spread in Clusters 5 and 6. Explosive cyclones, highlighted in Figure 2, can be found in Clusters 1 and 2, and one case in Cluster
5, indicating a preference for these types of PV structures. In fact, most of the cyclones (55%) are already in their weakening
phase when they enter the domain, as also indicated by the clustering of minimum SLP locations near the 27E meridian. This
suggests that for a significant number of systems, the mature phase of the cyclones occurs upstream of the domain boundary,
and they affect the eastern Mediterranean domain at their decay phase. The results also highlight how surface SLP features -
200  such as the location and depth of minima - correspond closely to upper-level PV structures, particularly the positioning of the
PV troughs and ridges that guide cyclogenesis in this region.

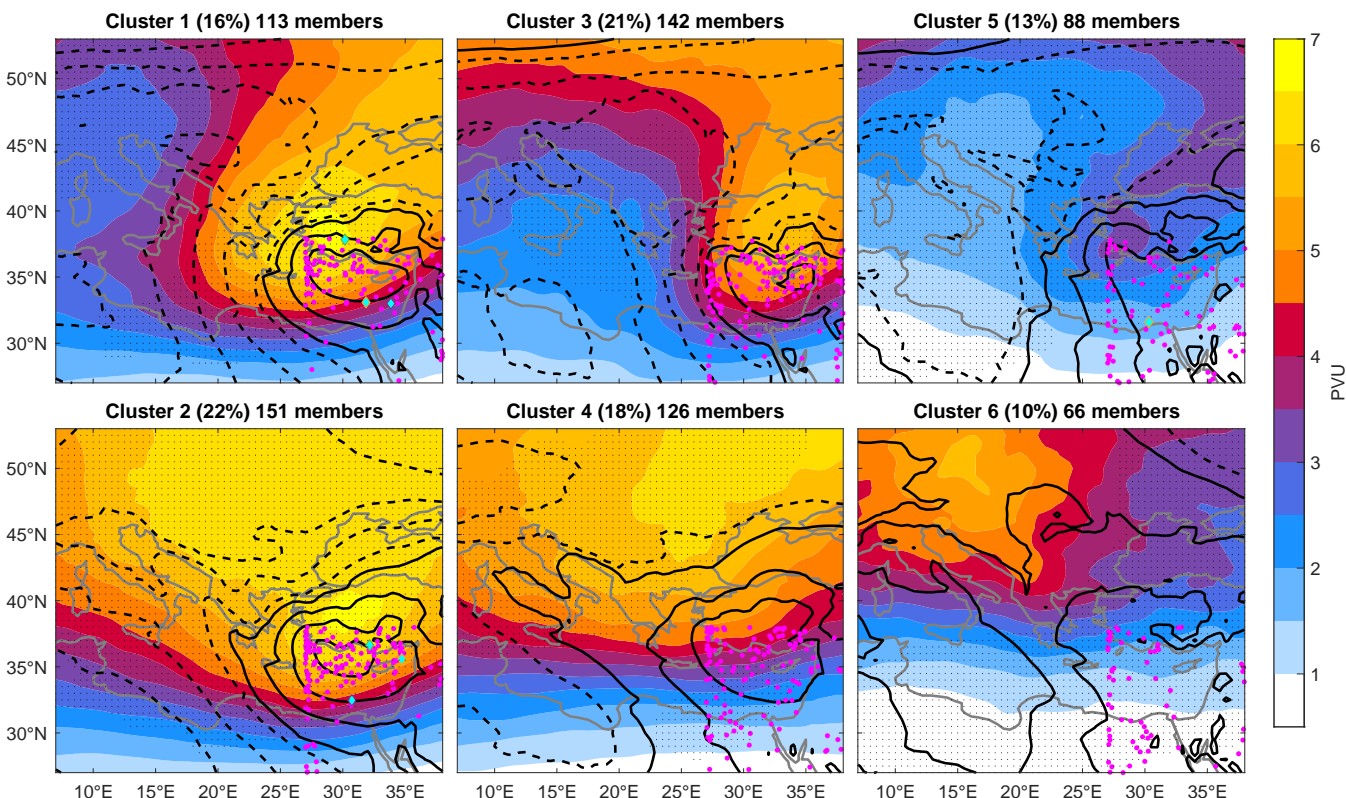

**Figure 2.** Composite mean of upper-level (320-340 K mean) PV (PVU, shading) and SLP (black contours at 2-hPa intervals, dashed over
1015 hPa) at minimum SLP time along the track within the study domain (magenta dots, blue diamonds marking explosive cyclones). Black
hatching indicates a 99% significance level (using the Gridded Student's T-test) of the PV field concerning the total cyclone average. The
number of cyclones and their percentage out of the total are indicated in the title of each cluster panel. (see text for details)

### 3.3  Seasonality

While regional cyclones occur mostly during the extended winter season (Figure 1), the SOM classification highlights a more
refined and distinguished seasonal signature for each of the resulting clusters. In January and December, there is a higher





frequency of Clusters 1 and 3 (Figure 3), indicating that these atmospheric patterns are more prevalent at the onset of winter. This is consistent with the typical intensification of upper-level troughs and stronger baroclinic activity during winter, favouring the development of pronounced PV anomalies. In contrast, Clusters 2 and 4 primarily comprise the late winter, with Cluster 2 occurring mostly between January to March and early spring, while Cluster 4 accounts for the months of March and April. This suggests that there is a shift in the regional dynamical cyclone lifecycles as winter progresses. As expected by their overall

lower PV, Clusters 5 and 6 appear mostly during the transitional seasons of autumn and spring (respectively). These clusters appear even in summer, but with lower frequencies, highlighting their association with different atmospheric conditions typical of these periods. These findings highlight the seasonal variability of the PV patterns and underlying dynamical changes, with specific clusters associated with distinct phases of the seasonal cycle. At the same time, these results suggest that a seasonal distinction alone would not resolve the full variability of cyclones, as most clusters also occur outside of their typical season,

and different cyclone types occur at any given month.

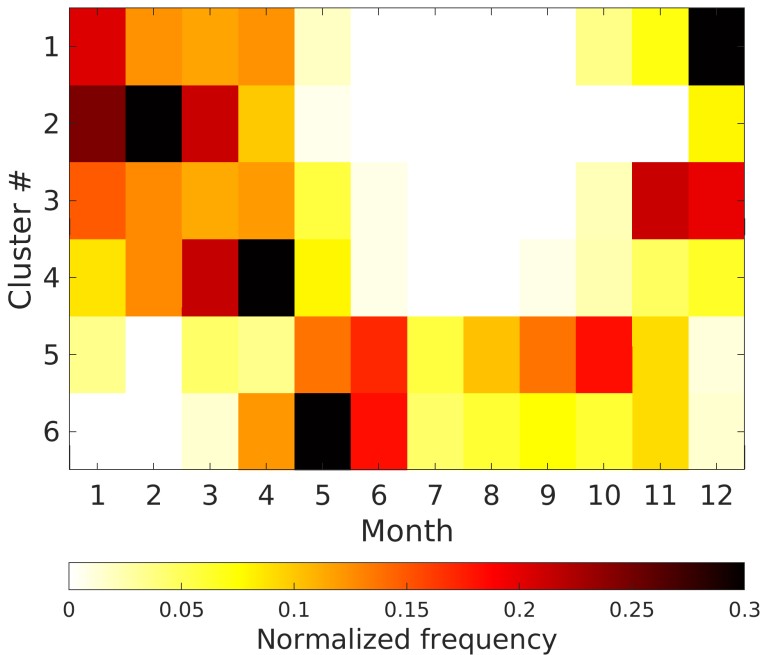

**Figure 3.** Monthly distribution of cyclone occurrences by cluster. The relative frequency (shading) is normalised relative to the cluster size, i.e., each row sums up to one.

To further quantify the dynamical characteristics of each cluster, Table 1 summarises mean cyclone track properties, including deepening rate, mean minimum SLP, duration within the domain, and distance travelled both within the defined domain and along the entire cyclone track. Cluster 1 is characterised by a relatively high deepening rate and the longest duration, along with moderate distances travelled both within the domain (∼978 km) and overall (∼1402 km). This indicates long-lived,

dynamically active systems that tend to remain partly stationary during their time in the research domain, most likely benefiting from persistent baroclinic conditions. In contrast, Cluster 2 exhibits a lower deepening rate and shorter duration but



travels significant distances within (∼927 km) and outside the domain (∼1794 km). These cyclones, although not particularly intense, show greater mobility and traverse large geographical extents. Cyclones in Cluster 3 display the highest deepening rate, although this cluster has no explosive cyclones. They have a moderate duration and moderate total distance (∼1404 km)

but the longest distance within the domain (∼1022 km). Cluster 4 contains the shortest-lived and weakest cyclones (lowest deepening rate), travelling shorter distances inside the domain (∼677 km) yet covering the largest total distances (∼1852 km). This indicates transient systems that quickly pass through the domain without significant intensification but continue travelling extensively to the East of the domain.

Clusters 5 and 6, typical of transition seasons and summer, reflect characteristics of thermally driven or weakly baroclinic

cyclones like "Sharav" lows, as will be shown in Section 3.5. The cyclones in Cluster 5 show moderate deepening rates and duration, with moderate distances travelled inside (∼911 km) but the shortest overall distances (∼1312 km), consistent with relatively stationary cyclones influenced by weak upper-level forcing and local thermal gradients. Conversely, cyclones in Cluster 6 exhibit the lowest deepening rate and short durations but relatively large overall distances (∼1722 km). These features suggest weaker systems, likely influenced by transient upper-level disturbances or weaker baroclinicity, yet displaying

notable mobility beyond the study domain. These characteristics align well with the seasonal variability observed in cyclone behaviour in the region.

| Cluster | Deepening rate [hPa/12h] | Mean Min SLP [hPa] | Duration [h] | Distance in the domain [km] | Overall Distance [km] |
|---------|--------------------------|--------------------|--------------|-----------------------------|-----------------------|
| 1 | $0.20 \pm 0.03$ | $1008.6 \pm 7.25$ | $41.84 \pm 2.62$ | $978.38 \pm 81.49$ | $1402.10 \pm 75.52$ |
| 2 | $0.14 \pm 0.02$ | $1007.2 \pm 6.84$ | $29.46 \pm 1.77$ | $927.31 \pm 85.18$ | $1794.20 \pm 70.50$ |
| 3 | $0.23 \pm 0.02$ | $1009.7 \pm 7.67$ | $33.44 \pm 1.99$ | $1022.10 \pm 85.80$ | $1403.80 \pm 65.72$ |
| 4 | $0.10 \pm 0.02$ | $1008.0 \pm 6.05$ | $21.39 \pm 1.62$ | $676.55 \pm 77.10$ | $1851.60 \pm 83.21$ |
| 5 | $0.15 \pm 0.02$ | $1007.0 \pm 4.69$ | $33.98 \pm 2.67$ | $910.87 \pm 97.40$ | $1311.90 \pm 88.84$ |
| 6 | $0.06 \pm 0.02$ | $1007.0 \pm 4.67$ | $23.64 \pm 3.31$ | $743.83 \pm 110.43$ | $1722.30 \pm 122.78$ |

**Table 1.** Mean and standard deviation values of key track characteristics for each cyclone cluster. The values include the deepening rate over 12 hours (Bergeron), minimum SLP along the track (hPa), duration of the cyclone within the study domain (h), and total distance travelled (km), both inside the domain and overall the life-cycle of the cyclone.

### 3.3.1   Comparison of ERA5 and IMERG Precipitation

We first compare precipitation from ERA5 and IMERG, by cluster, to assess the suitability of ERA5 precipitation forecasts for the long-term climatological comparison among clusters. Examining composites of 1-hourly precipitation at peak cyclone

intensity by cluster (Figure 4) allows a comparison in terms of geographical features and the intensity of precipitation hotspots. Generally, both the geographical distribution of precipitation and the differences among clusters agree well when comparing the IMERG and ERA5 data. The higher resolution of IMERG allows for more detailed structures of precipitation and more intense maxima compared to ERA5. Thus, while ERA5 is useful for comparative climatological studies, one should note that





ERA5 precipitation likely misses some geographical hotspots, crucial for assessing surface impacts, especially over land near
coastal regions.

Further examination of the precipitation composites reveals large differences among EMC clusters. Precipitation in the winter clusters 1-4 peaks along the east coast of the Mediterranean basin and in the lee of the Turkish mountains. Although the geographic location and pattern of precipitation impacts are similar, the intensity varies considerably between clusters. Cluster 1 shows a broad precipitation signature across the entire region, especially in the northern part of the domain, south of Turkey,
as well as a hotspot near the eastern coastal areas, and over the Mediterranean Sea north of Egypt. Cluster 2 also has a broad precipitation signature reaching southern areas, but it is slightly less intense than Cluster 1. Cluster 3 presents a more limited precipitation signature compared to the previous clusters, without major precipitation along the Turkish coasts. Conversely, precipitation in Cluster 4 mainly occurs along the Turkish coast. The latter differences emerge from the very different PV signature in those clusters, with a tilted PV streamer in Cluster 3, fixing precipitation ahead near the eastern coast, compared
to a zonal PV orientation in Cluster 4, with precipitation spread along the southern coast of Turkey (Figure 2).

Clusters 5 and 6, corresponding to the transitional seasons and summer, exhibit different precipitation patterns and are much drier on average. Cluster 5 exhibits rain signatures primarily over the sea and southern Turkey, with weaker intensities compared to the winter-associated clusters. Cluster 6 has the weakest precipitation signature overall, with a few notable areas that are mainly visible in the IMERG data. Precipitation in ERA5 is predominantly convective, indicating that model precipitation in
the region is highly sensitive to the choice of parameterisation, especially over the sea and coastal regions. Yet, the distinction by clusters shows fundamental differences in the variability of precipitation, which are consistent across datasets. We thus employ ERA5 precipitation in the next subsections for its longer temporal coverage to examine precipitation climatologies across EMC types.





**Figure 4.** 1-h precipitation composite (mm/h, shading) for ERA5 (top panels for each cluster (a-c,g-i)) and IMERG (bottom panels for each cluster (d-f,j-l)) during the hour ending at minimum SLP time, for the years 2000-2017, covered by both datasets. SLP is shown in black contours at 2-hPa intervals (dashed over 1015 hPa). In the ERA5 panels, light blue and magenta dots mark grid points dominated by (more than 60% of the total precipitation) convective precipitation or large-scale precipitation, respectively.




### 3.3.2 ERA5 3-day accumulated precipitation

To account for sustained precipitation during the entire event of EMC passage, we next examine the 3-day accumulation of precipitation, centred around the minimum SLP time. The results reveal significant differences in precipitation hazards among the clusters (Figure 5). The winter clusters contribute the most to the overall precipitation, with maxima to the east of the mean minimum SLP centre. In particular, Cluster 1 exhibits the most intense composite mean precipitation, with prominent maxima near the eastern coastal areas exceeding 42 mm. Cluster 2 also shows considerable precipitation, with a notable

peak to the east of the mean minimum SLP centre. In contrast, the transition-seasons Clusters 5 and 6 are characterised by much drier conditions, with only 5 mm along the eastern coast and $\sim 10$ mm near the northern coasts. Clusters 3 and 4 show an intermediate situation, with more limited precipitation signatures compared to Clusters 1 and 2. Similar to the 1-h precipitation, the precipitation in Cluster 4 is concentrated along the northern part of the region, compared to Cluster 3, where precipitation extends along the eastern coast of the Mediterranean.

Figure 5 indicates that most precipitation associated with cyclone events in this area is of convective scale (cyan dots in Figure 5), with large-scale precipitation (magenta dots in Figure 5) dominating only over southern Turkey in the winter-dominant Clusters 1–3. This agrees with Portal et al. (2025) and Raveh-Rubin and Wernli (2015), who found that convective precipitation is generally uncommon across the Mediterranean during winter. In contrast, the persistent dominance of convective precipitation in the southern and eastern parts of the domain, even in winter, highlights the distinctive behaviour of Eastern

Mediterranean cyclones. Moreover, even when increasing the threshold to 80%, most grid points, particularly those over the sea, remain dominated by convective precipitation, underscoring the robustness of this signal.

During the transition seasons and summer, the results remain consistent with previous findings (Portal et al., 2025). Cluster 5, which shows a strong autumnal signal and is linked to Red Sea Trough systems, exhibits convective precipitation patterns characteristic of active Red Sea Troughs (de Vries et al., 2013; Krichak et al., 2012; Ziv et al., 2022).

Cluster 6, on the other hand, shows a predominance of large-scale precipitation in the southern part of the domain, possibly enhanced by tropical moisture sources (Ciric et al., 2018; Rubin et al., 2007). The dominance of ERA5 convective precipitation, stemming from the convective parametrisation, further suggests potentially high model uncertainty during these events.

The fact that Cluster 1 has more precipitation than Cluster 2 in the Eastern Mediterranean region is particularly interesting. Although Cluster 1 EMCs are, on average, shallower and with lower upper-tropospheric PV over the eastern Mediterranean,

the accumulated precipitation over three days is comparatively higher by up to 10 mm in most regions. This highlights the importance of the regional distribution of PV, controlled by the life-cycle of the baroclinic Rossby wave. In Cluster 1, a clear signature of anticyclonic Rossby wave breaking (AWB) is prevalent, downstream of an enhanced ridge. This large-scale configuration supports different surface cyclone lifecycle and duration. Indeed, the average duration of cyclones in Cluster 1 is more than 50% longer than in Cluster 2 (Table 1), and the mean deepening rate is stronger. This implies that Cluster 1

is associated with more stagnant and rapidly deepening EMCs than Cluster 2, controlled by the large-scale wave-breaking configuration. This prolonged duration can result in more precipitation when accumulating over longer periods. This analysis aligns with common knowledge among forecasters in this region (Goldreich, 2003; Zangvil and Druian, 1990), namely, a trough




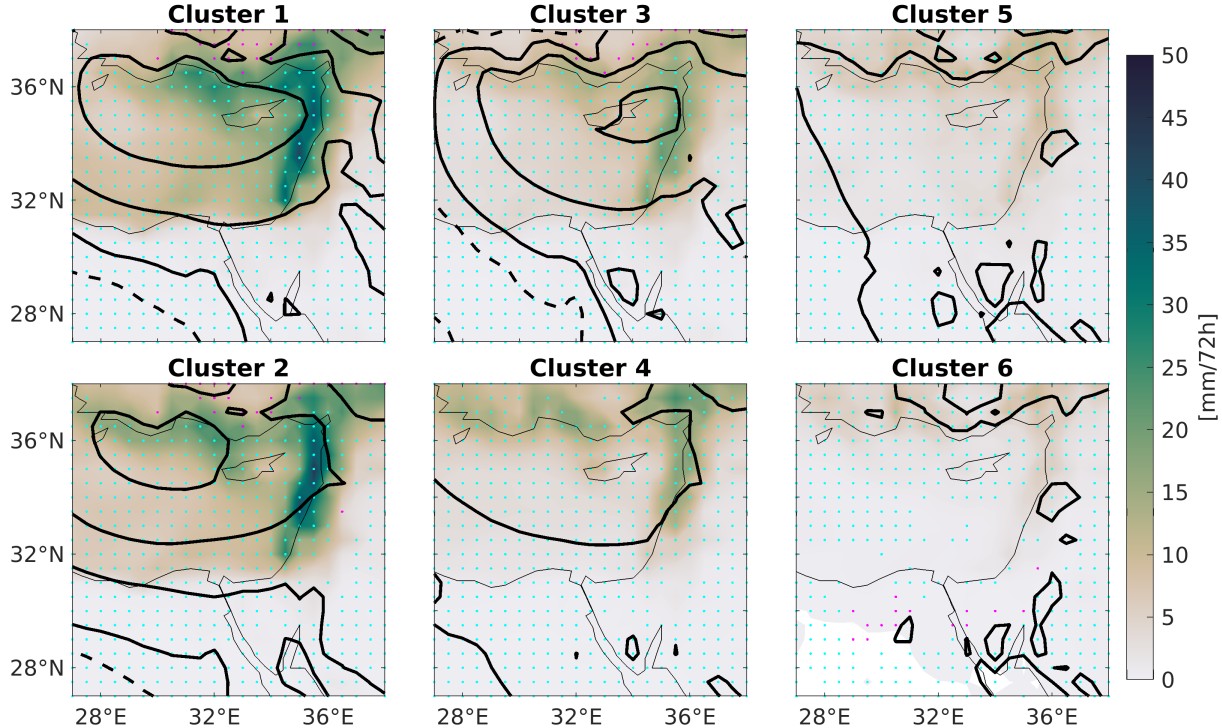

**Figure 5.** Composite mean 3-day ERA5 accumulated precipitation (shaded, mm/72h), beginning at 6 am of the day of minimum cyclone SLP, by cluster. Cyan-dotted areas indicate regions where ERA5 convective-scale precipitation exceeds 60% and magenta-dotted areas where ERA5 large-scale precipitation exceeds 60%. SLP is shown with black contours at 2-hPa intervals, with dashed contours for values above 1015 hPa. Note the smaller domain compared to Figure 2, covering only the cyclone-detection region.

extending westward often leads to a greater precipitation impact. A similar AWB trough structure also appears in Cluster 3 and exhibits similar seasonality. However, despite these similarities, it is evident that cyclones in Cluster 3 produce considerably
less precipitation compared to Cluster 1 cyclones. This difference can potentially be explained by the weaker upper-level PV and PV gradients observed in Cluster 3 compared to Cluster 1. Along the same lines, Clusters 2 and 4 also exhibit similarities in the shape and distribution of upper-level PV. Nevertheless, Cluster 4 is associated with significantly lower precipitation amounts compared to Cluster 2, as well as relatively lower PV and PV gradients. However, in this case, Cluster 4 occurrence is shifted towards later winter and spring compared to Cluster 2, explaining its lower PV. In summary, the composite analyses
reinforce the inherent relationship between the spatial distribution and gradients of upper-level PV and surface precipitation impacts.

### 3.4 Precipitation distribution, variability and extremes

It is important to consider the distribution of the accumulated precipitation that constructs the spatial composite mean patterns. While the mean provides a view of most cases, it may obscure information about extremes, their number and location.




Moreover, relying solely on the mean values does not indicate whether these values stem from a few intense events or if the precipitation distribution is more evenly spread across the cases. Therefore, for each of the 686 cyclones we examined the distribution of ERA5 3-day accumulated precipitation within the research domain and cluster type (Figure 6). Clusters 1, 2, 3 and 4 are characterised by a relatively uniform distribution, indicating that precipitation events within these clusters consistently cover a wide range of intensities, and there are no isolated heavy rainfall events that dominate the distribution. Conversely,

other clusters are primarily influenced by a few individual events contributing substantial rainfall amounts. Cluster 5, for instance, includes many events with little or no precipitation and a long tail of extremes, while Cluster 6 features consistently low precipitation values throughout. This analysis also highlights the significant impact that explosive cyclones (marked with "b" in Figure 6) can have in the region, particularly those that lie on the extreme end of the precipitation distribution. For example, Cluster 5, which is generally characterised as a drier cluster, includes only one explosive cyclone. This single event,

which occurred in 12-13 March 2020, contributed disproportionately to the mean precipitation associated with the cluster. Understanding this distribution is crucial, as it reveals whether the observed mean values are driven by frequent but less intense events or by rare yet highly impactful cyclones. This distinction is essential for accurately assessing the overall influence of cyclones on regional precipitation patterns.

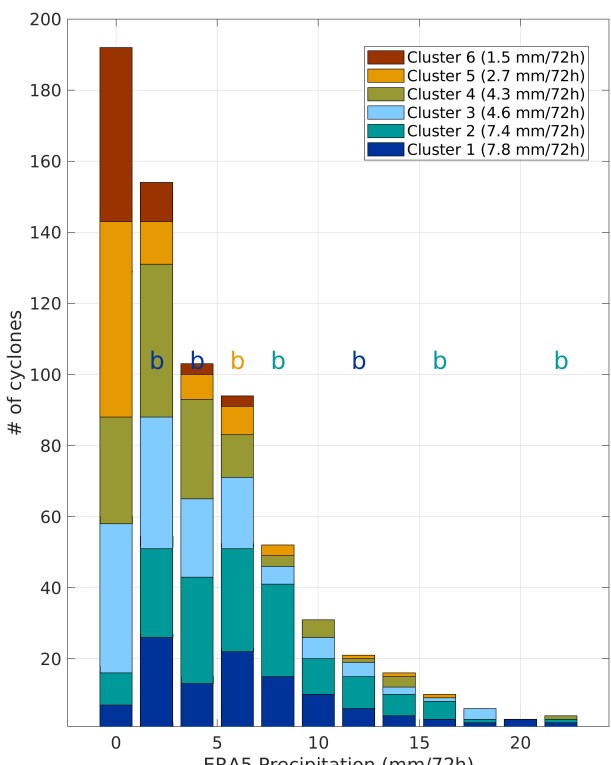

**Figure 6.** The distribution of mean 3-day ERA5 accumulated precipitation within the domain, per cyclone. Bars are grouped by precipitation bins and colour-coded by cluster, with the legend indicating the cluster number and its overall mean grid-point precipitation (mm/72h). The letter b marks explosive cyclones (Bergeron in 12 hours $\geq 1$) in their corresponding bins, colored by cluster.





To focus on the tail of the distribution and locally measured precipitation, we examine the most extreme 3-day precipitation
in each station in Israel. Figure 7 presents, for each station and for each non-summer season, the clusters associated with its
highest 72-hour accumulated precipitation events during cyclone days. Summer is excluded due to negligible rainfall in Israel
during these months.

Consistent with their mean precipitation, during winter months (DJF) Clusters 1 and 2 dominate the most extreme events.
This aligns with their pronounced upper-level PV structures and their association with dynamically intense cyclones. In some
cases, these events result in over ∼300 mm of accumulated rainfall at a single station over a three-day period. In contrast,
the transitional seasons (MAM and SON) exhibit much greater variability, with extreme events distributed more evenly across
different clusters. Especially, Cluster 5 extremes dominate southern Israel in spring, and some coastal stations in autumn,
despite having low mean precipitation signatures overall. Thus, even in a confined region, extreme precipitation is not typically
driven by a single dominant synoptic pattern, but can arise from a broader range of atmospheric conditions.

In most cases, each station's extreme event is linked to a specific cyclone, rather than the same system simultaneously af-
fecting many stations. Table 2 shows the number of cyclones contributing to each of the seasons' maximal events. There is no
dominant cyclone for all the stations together. This highlights the localised or regionally confined nature of most extreme pre-
cipitation events, even in the presence of dynamically intense cyclones. Only a few cyclones were associated with widespread
impacts, the most notable affecting up to 18 stations (not shown), while most contributed to extremes at just 1 to 5 stations,
underscoring the predominance of spatially limited, convective extreme events.

One should note, however, that a broader influence of certain cyclones becomes evident when considering also the second
and third-highest precipitation events per station (Table 2). Many cyclones that appear as second or third-ranked events in
some stations are the same ones that produced the most extreme rainfall at other locations. This suggests that while individual
cyclones may not simultaneously cause peak rainfall across many stations, they can still impact a wide region with varying
intensities.





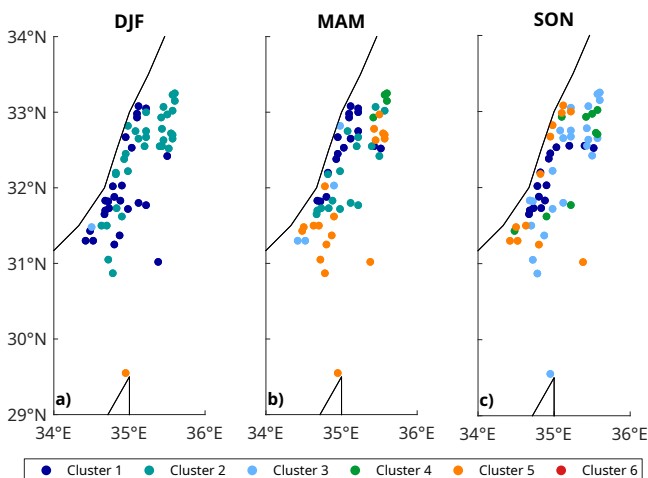

**Figure 7.** Clusters associated with the top 3-day accumulated precipitation event at each IMS rain gauge station, shown separately for winter (DJF panel a), spring (MAM panel b), and autumn (SON panel c) and colour coded by cluster.

| Season | rank-1 | rank-2 | rank-3 |
|--------|--------|--------|--------|
| DJF | 17 | 26 | 31 |
| MAM | 13 | 17 | 17 |
| SON | 13 | 18 | 18 |

**Table 2.** Number of cyclones contributing to the top-ranked 3-day precipitation events (1, 2, and 3) for each season. The numbers range between 1 (one cyclone produces an extreme in all 60 stations), and 60 (each cyclone produces an extreme in a single station).

## 3.5 Temperature anomalies and extremes

To further understand surface impacts of cyclones, we examine the temperature anomalies associated with the different PV patterns. Figure 8 presents the composite mean 2-m temperature anomalies calculated from the monthly mean climatology for each cluster. Overlaid are the grid points where more than 15% of the cyclones experienced extreme cold or warm daily 350 temperature anomalies (below the 5th or above the 95th extreme percentile of the distribution, respectively). As either cold or warm anomalies are expected with a potential frontal passage, composites are shown for the time of minimum SLP and for 12 hours after.

At the time of minimum SLP, all clusters except for Cluster 3 exhibit positive anomalies in the eastern part of the domain, with a strong signature over land areas, concurrent with the warm sector of the cyclone. Clusters 4 and 6, in particular, demon-355 strate extremely hot temperatures in the southern part of the domain. These hot extremes are known as Sharav heat lows in the region, conductive for warm, dry, and often dusty extremes in spring (Alpert and Ziv, 1989; Dayan et al., 2008; Nissenbaum et al., 2023). At the same time, over the sea, all clusters but Cluster 6 exhibit cold anomalies. Winter Clusters 1 and 2 exhibit



strong cold anomalies (compared to their respective months). In these clusters extremely cold temperatures are seen to the west and south of the cyclone centre.

Upon the passage of the cyclones, 12 hours after peak intensity, most of the domain becomes colder (Figure 8). In Clusters 1 and 2, the warm sector associated with the cyclones traverses the domain, leading to anomalously cold temperatures throughout the region. Cluster 3, which did not exhibit any warm anomalies, shows the domain becoming even colder (at t=+12), especially over land; this may be attributed to the upper-level PV pattern indicating strong air advection from the north. For Clusters 4 to 6, while the warm anomalies over land persist, their intensity diminishes. Notably, Clusters 4 and 6 maintain pronounced

warm extremes over land. This analysis highlights the frontal behaviour of cyclones associated with winter seasons and the presence of a warm core in cyclones more typical of late spring and summer, particularly evident in Clusters 4 and 6. This cluster-specific analysis highlights the strong connection between upper-level PV patterns and surface temperature anomalies and extremes.

It should be noted, however, that the timing of the minimum SLP may influence the results. While the distribution of

minimum SLP times is generally uniform, certain timings are somewhat favoured and could therefore affect the composite mean, particularly when comparing differences between two specific time steps. To assess the robustness of our findings, we also examined temperature differences using a 24-h interval. Although the magnitude of the anomalies is weaker, the overall patterns and behaviour remained consistent.

The results underscore that, on average, cyclones introduce temperature anomalies and often extremes to the region. This

finding emphasises the importance of understanding the link between the upper-level atmospheric state and its impact near the surface, which could not be implied by the SLP pattern alone.




**Figure 8.** Composite mean 2-m temperature anomalies, shown at the time of min SLP (top panel of each cluster (a-c,g-i)) and 12 hours after (bottom panel of each cluster (d-f,j-l)). Anomalies are shown as a difference from the monthly mean climatology (K, shading). Composite mean SLP is shown for each time (black contours at 2-hPa intervals dashed over 1015 hPa). The dots mark grid points where more than 15% of the cyclones induce a warm/cold extreme (within the upper/lower local 5%, respectively).





## 3.6 Observed cyclone frequency trends

We here examine the inter-annual frequency of EMCs in each cluster. As different types of cyclones have varying impacts on the surface, their respective trend may suggest expected surface changes in the region in the future climate. When considering all clusters together, there is no significant trend in EMC frequency (Figure 1). However, the examination of specific clusters reveals that some of the clusters exhibit significant and opposing trends in the observed period (Figure 9). Clusters 5 and 6 demonstrate a significant increase, while a decrease is seen in Cluster 4. Other clusters do not display significant trends. The significance of the trend for Cluster 5 is the strongest, increasing during the period from 1 to 3 cyclones per year on average.

The compensating trends of the different clusters are especially meaningful when taking into account the surface impacts of each cyclone type. As Cluster 5 is associated with dry and warm anomalies with sporadic precipitation extremes, a potential shift toward drier and warmer conditions under cyclones is likely to occur if this trend continues. Thus, considering cyclone frequency as a whole is not a sufficient proxy for precipitation trends, and cluster-specific information is key for understanding long-term changes in precipitation patterns, affecting water resources, agriculture, and the health of the ecosystem in the region.

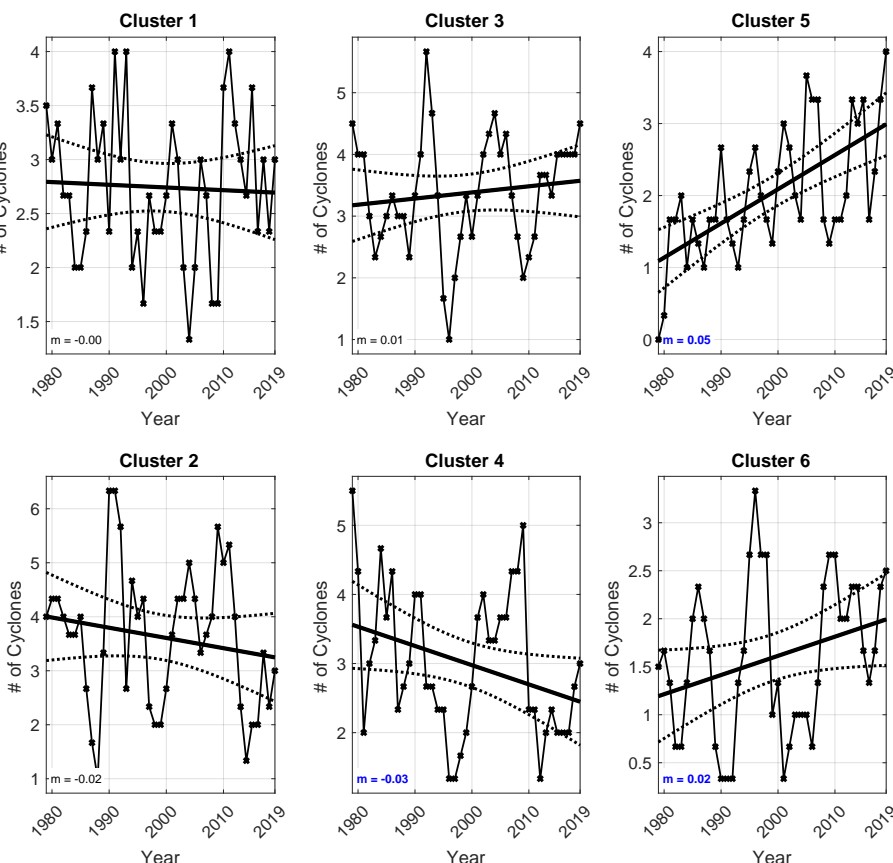

**Figure 9.** As in fig. 1a, but shown separately for each cluster. The slope of each linear trend is annotated as m (cyclones per year). Trends significant at the 90% confidence level (Mann-Kendall test, $\alpha = 0.1$) are indicated by blue coloured m.





## 4   Discussion and Conclusions

This study aimed to understand what drives the variability of Eastern Mediterranean cyclones and their associated surface weather, including extremes. To this end, we classified Eastern Mediterranean cyclones according to the regional PV distribution using a self-organising map algorithm, resulting in six distinct EMC types, or clusters (Figure 10):

- **Cluster 1:** Characterised by anticyclonic wave breaking with high positive PV values over the region and amplified upstream ridge over central Europe. These early-winter cyclones are relatively deep with long duration inside the do-

main, and on average, relatively high deepening rates. Their thermal structure shows strong frontal characteristics with pronounced warm and cold anomalies. Cluster 1 contributes the most to annual precipitation.

- **Cluster 2:** Defined by a wide trough with high PV values, typically occurring in late winter. This cluster includes the deepest cyclones in the climatological cyclogenesis zone near Cyprus. Like Cluster 1, it shows strong frontal behaviour and contributes significantly to annual precipitation.

- **Cluster 3:** Associated with anticyclonic wave breaking and lower PV values than Cluster 1, also typical of early winter. Cyclones in this cluster are shallower, with colder-than-normal temperatures and moderate precipitation impacts compared to Clusters 1 and 2, but they have the highest average deepening rate and travel the longest distance within the domain.

- **Cluster 4:** Exhibits a zonal PV structure with a more poleward tropopause and lower PV than Cluster 2. It occurs

primarily in late winter and spring. Its warm-core signature and thermal structure is associated with weaker precipitation. The cyclones in the cluster have the shortest mean duration and distance within the domain.

- **Cluster 5:** Characterised by low PV values in the region, with hints of anticyclonic wave breaking. It occurs mostly in spring, autumn, and summer, with distinct peaks in June and October. This cluster is generally dry, with a few extreme events contributing most of its precipitation. Interestingly, the frequency of Cluster 5 cyclones significantly increases. If

this trend continues, and since no significant trend exists for overall EMC numbers, this may suggest warmer and drier cyclone surface weather in the region, with potentially sporadic extreme events.

- **Cluster 6:** Defined by low PV values over the region and a high-PV signature northwest of the area. It is mainly active in spring and summer and is associated with strong warm anomalies, and has the lowest averaged deepening rate.

Together, these six cyclone types capture the main modes of variability in EMC structure and evolution, each associated

with distinct seasonal and dynamical characteristics. The detailed classification of cyclones into distinct clusters based on upper-level PV patterns provides a more nuanced understanding of cyclone behaviour and its surface impacts. By identifying specific cyclone types and their associated effects on precipitation and temperature, our study offers valuable insights for improving regional climate models and forecasting systems. Importantly, the use of large-scale PV structures as the basis for classification has a practical advantage: these upper-level features are more easily detected and more reliably simulated in





**Figure 10.** Schematic summary of the cyclone clusters. For each cluster, the upper panels show the large classification domain with the 3-PVU surface and SLP. The lower panels present a zoomed view over the Eastern Mediterranean, displaying temperature anomalies and accumulated precipitation (illustration by Itai Raveh).

models compared to localised variables such as convective precipitation extremes. Therefore, understanding cyclone behaviour through PV patterns not only offers physical insight but also provides a robust framework that can support improvements in future model performance and predictive skill under changing climate conditions.

While these findings provide valuable insights, several limitations should be noted. (i) The analysis was conducted within a fixed geographical domain, which can lead to noisier mean composites due to varying cyclone positions. Yet, it enables the

accurate assessment of the cyclones' real impacts and their interaction with regional features such as coastlines and topography. (ii) The trend estimates are based on a relatively small number of events, which limits their statistical robustness. The detected trends should therefore be interpreted as potential emerging signals rather than definitive long-term changes. Using a lower confidence level for the tracking algorithm could increase the sample size and strengthen significance, but at the cost of reduced





reliability. (iii) Finally, the use of a SOM as a classification tool also introduces uncertainty, as it is based solely on statistical
similarity rather than physical constraints, potentially smoothing out case-specific details. Nevertheless, this approach provides
a more objective and reproducible framework for identifying recurring cyclone patterns compared with manual or subjective
classifications.

Our study advances earlier efforts to classify EMCs by employing an objective and data-driven approach. Previous studies
e.g., Alpert et al. (2004) used semi-objective classification methods relying on expert judgment to categorise regional synoptic
systems into detailed classes based on SLP maps. While effective, this approach is subjective and labour-intensive. More recent
studies, e.g., Flocas et al. (2010), classified Eastern Mediterranean cyclone tracks using a single tracking algorithm, focusing
on track density and trends over a 40-year period. However, their classification relied on surface synoptic features and was
constrained by lower resolution and older datasets. In contrast, our study employs SOM to classify EMCs into six objective
clusters based on upper-level PV patterns during cyclones' peak intensity in the region. This automated method reduces subjec-
tivity, enhances reproducibility, and provides a process-based analysis of the relationship between atmospheric conditions and
surface impacts. By using state-of-the-art high-resolution data, our approach offers a more precise and accurate depiction of
cyclones, improving upon the limitations of older studies and delivering a more robust framework for understanding regional
climatic variability.

Our analysis suggests a possible trend towards a higher frequency of cyclones with drier impacts, consistent with projections
in Reale et al. (2022). The Med-CORDEX ensemble of climate models indicates that future drier conditions in the Eastern
Mediterranean could be influenced by both a decrease in the number of cyclones and a reduction in the intensity of rainy
events. Our findings suggest an increase in dry cyclones, together with no change in the overall number of cyclones, which
might already contribute to these anticipated changes. If the observed increase in dry cyclones together with sporadic extremes
continues, it will have implications for regional climate and water resources. Therefore, incorporating these possible changes
into future climate adaptation strategies is key for the evaluation of future trends.

Building on these findings, future work should further refine and extend the classification approach. To enhance our under-
standing of cyclonic behaviour and its impacts, it is crucial to continuously update cyclone classification methods with new
data and apply them to future simulated climate scenarios. This approach would offer insights into how cyclone characteristics
and more diverse forms of surface impacts (including compound events, air quality and others) may change under varying
climate conditions, aiding in the development of more effective adaptation and mitigation strategies.

*Author contributions.* TSG led the research, analysed and visualised the data, and wrote the first draft of the manuscript. SRR conceptualised
the research and acquired funding. LMR and SRR contributed input and supervised the research. All authors contributed to revising the
manuscript to its final form.

*Competing interests.* SRR is a member of the editorial board of Weather and Climate Dynamics.



460 *Acknowledgements.* This work was funded by the Israel Science Foundation (grant no. 1242/23) and the De Botton Centre for Marine Science at the Weizmann Institute of Science. This work contributes to the Med-World and Tuning for Deserts Consortia, funded by the Council for Higher Education in Israel. We thank Itai Raveh (Weizmann Institute) for helping with the production of Figure 10. This work benefited from language suggestions provided by ChatGPT and Grammarly.



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
