# Peer review of "A Climatological Perspective on Cyclones and Surface Impacts in the Eastern Mediterranean Using Potential Vorticity-Based Classification"

_EGUsphere, 2025_

## Referee Comment (RC1)

**Review of *A Climatological Perspective on Cyclones and Surface Impacts in the Eastern Mediterranean Using Potential Vorticity-Based Classification* by Gens et al.**

In this work, Eastern Mediterranean cyclones are separated into six classes based on their upper-level PV patterns over the region. The resulting classes reveal a clear link between upper-level dynamics, seasonality, and cyclone impacts in terms of mean and extreme precipitation and temperature. The study presents a novel and useful categorisation of Eastern Mediterranean cyclones, is well constructed, and is firmly grounded in the existing literature on the topic. I recommend publication in *Weather and Climate Dynamics* after the following minor revisions are addressed.

**General Comments**

Abstract: The abstract would benefit from shortening and simplification. It is relatively long and contains several lengthy, at times convoluted, sentences that could be improved in terms of clarity and readability.

Figure 6: I question the analysis of extremes based on precipitation accumulated and then averaged over such a wide region. In particular when convection plays a role, local precipitation totals can be very high despite relatively modest values of spatial averages over the entire domain. A discussion of this limitation, or an alternative metric that better captures local extremes, would strengthen the analysis. For example, I would suggest applying a similar analysis as this to ERA5 precipitation data, taken by grid-point or aggregated over small (land) regions.

Figure 7 : What is the range of top values in the different stations ? You hint to this in the text in lines 330, but I would appreciated a few more details about the distribution. Also, could you compare these results with a similar analysis applied to ERA5 precipitation data ?

**Specific comments**

Line 33 : « high » instead of « tremendous ». Generally, I would recommend to limit the use of the word « tremendous » in a scientific context.

Line 39 : You can add ref to Chiericoni et al. 2025 [1].

Lines 93-101 : I suggest using present tense, as in the rest of the manuscript.

Line 100 : « outside the 5th or 95th percentile range ».

Line 135 : Is the choice of six clusters subjectively based on the clustering outcome, or is it based on some metrics as in Givon et al. 2024 (described in Appendix A) ?

Line 164 : I understand what you mean with this sentence, but I find the term « rainy seasons » confusing. Consider rephrasing with the following : « The analysis uses annual periods running from 1 August to 31 July instead of calendar years. »

Line 175 : I would suggest « support » instead of « substantiate ».

Line 244 : You can refer to [2] to support your statement.

Lines 277-278: Portal et al. 2025, Figure S4 also shows that lightning frequency (i.e., a proxy of deep convection) conditioned on the presence of cyclones is high in the north-eastern Mediterranean in winter. This agrees with the findings described in the following sentence (lines 278-280).

Line 298 : I would suggest replacing « a trough extending westward » with « a south-west–to–north-east–tilted trough ».

Line 303 : Remove « however ».

Line 390-392 : I suggest the use of present tense.

**References**

[1] Chiericoni, M., Fosser, G., Flaounas, E. *et al.* Unravelling drivers of the future Mediterranean precipitation paradox during cyclones. *npj Clim Atmos Sci* **8**, 260 (2025). https://doi.org/10.1038/s41612-025-01121-w

[2] Lavers, D. A., Simmons, A., Vamborg, F., and Rodwell, M. J.: An evaluation of ERA5 precipitation for climate monitoring, Q. J. Roy. Meteorol. Soc., 148, 3152–3165, 2022.

---

## Referee Comment (RC2)

Manuscript egusphere-2025-5979

A Climatological Perspective on Cyclones and Surface Impacts in the Eastern Mediterranean Using Potential Vorticity-Based Classification

Reviewer Decision

The manuscript should be reconsidered after major revisions.

Reviewer Summary/Narrative

The manuscript aims to diagnose different flavors of Eastern Mediterranean Cyclones (EMCs) by utilizing the self-organizing map (SOM) algorithm. The authors selected a 2 by 3 SOM with 6 nodes to group the EMCs based on PV data. Cyclone trends over time are discussed for the entire dataset as well as for each SOM node. Each node experiences the highest frequency of occurrence during different months throughout the year. Additionally, the different EMC groups have different deepening rates, duration, and distance associated with them. ERA5 precipitation was compared with IMERG data to demonstrate that the ERA5 precipitation is robust for both large-scale and convective precipitation. Additionally, precipitation by SOM node for observation sites in Israel was examined. The authors do a good job of relating the different SOM node PV patterns to precipitation locations and temperature changes. The end of the manuscript the highlights and importance of each node.

Based on the volume of major/minor concerns listed below, I recommend the manuscript should be reconsidered after major revisions.

Major Comments/Concerns

1. The authors describe the self-organizing map algorithm as an *objective* tool to group data, but this is incorrect. The user must specify the number of nodes, specify the shape of the map, the distance function used, and other metrics. This study would benefit from a more in-depth discussion about how the SOM algorithm works as well as quantitative metrics (quantization error, Sammon map, correlation of data to their assigned node, etc.) showing the final map chosen is a good fit for the PV data. Additionally, the labeling of the SOM nodes (not clusters) and using coordinate labeling should be used to be consistent with other studies. The authors also do not cite the SOM algorithm nor reference other studies to show that this is a viable and accepted method for grouping synoptic data.
2. More geographic specificity in the text and in figures would help clarify the results and orient the reader.
3. It is unclear why observational data for only Israel was used instead of everywhere within the study domain. Since this is only for a small portion of the domain, I think that observational data for the entire domain should be shown or this section should be omitted because it breaks the flow of the manuscript.
4. To better connect the PV SOM patterns to the precipitation, this study would benefit from including an additional figure (and small section) that provides additional

information about forcing mechanisms and moisture availability. A figure showing mid- or lower-level ascent and total column water vapor will aid the description about lee precipitation, "drier" nodes, large-scale vs convective precipitation. Some nodes were described as "drier," but I do not think that you can say that from precipitation alone. Rather, those nodes show a "lack of precipitation."

5. Given that the authors motivate this study by discussing how these EMCs cause many surface impacts, I believe that a discussion of the winds associated with these EMCs would be beneficial. If the authors only want to focus on precipitation and temperature changes, then that should be clearly stated in the introduction.

Specific Minor Comments and Line-by-line Edits

**ABSTRACT**

L2: recommend not using the word "significantly" unless using it for statistical reasons

L4: "EMC" should be "EMCs"

L7 & L20: "impacts" is often mentioned throughout the manuscript, but what are specific examples of these impacts?

L8 & throughout the manuscript: "Self-Organizing Maps" should not be capitalized ("self-organizing maps"). Also, this "maps" should not be plural. I would recommend saying "Using the self-organizing map algorithm to categorize ERA5 data into 6 distinct PV patterns that highlight different synoptic and precipitation patters."

L14-17: break this sentence into two sentences for easier readability

**INTRODUCTION**

L32: what is a compound event?

**2 DATA AND METHODS**

L89: add "by" in between "(SLP)" and "employing" and change ", and" to "to"

L94: why is the ERA5 data interpolated to 0.5 degree from the native 0.25 degree horizontal resolution?

L110: why only use observations from Israel? This should be explained earlier in the motivation of the study.

L121: more discussion about how the 10 cyclone-tracking differ would be beneficial. What are their strengths and weaknesses? What variables do they use to classify the cyclones?

L126: making Fig. 1b its own figure placed after this paragraph would introduce the domain better and make the manuscript flow better

**2.4 SELF-ORGANIZING MAP (SOM) CLASSIFICATION**

Subsection title: I would recommend changing "classification" to "algorithm"

L132-133: A reference to the original Kohonen 1982 algorithm should be added here. Acknowledging the MathWorks SOM package can be moved to the acknowledgment section. Additionally, more discussion about how the SOM algorithm spatially groups data should be added. To further justify why the SOM algorithm is an appropriate tool to use for this study, I would recommend referencing other synoptic studies that utilize this algorithm, such as Larson et al. 2025, Baiman et al. 2023, and LaChat et al. 2024. These articles are good examples of how to describe the algorithm.

Kohonen, T., 1982: Self-organized formation of topologically correct feature maps. Biol. Cybernetics, 43,59–69, https://doi.org/10.1007/BF00337288.

Larson, M. L., and A. C. Winters, 2025: A Climatology of Lee Cyclones across the Central United States, 1980–2021. Mon. Wea. Rev., 153, 2613–2633, https://doi.org/10.1175/MWR-D-25-0013.1.

Baiman, R., A. C. Winters, J. Lenaerts, and C. A. Shields, 2023: Synoptic drivers of atmospheric river induced precipitation near Dronning Maud Land, Antarctica. J. Geophys. Res. Atmos., 128, e2022JD037859, https://doi.org/10.1029/2022JD037859.

LaChat, G., K. A. Bowley, and M. Gervais, 2024: Diagnosing flavors of tropospheric Rossby wave breaking and their associated dynamical and sensible weather features. Mon. Wea. Rev., 152,513–530, https://doi.org/10.1175/MWR-D-23-0153.1.

L133: throughout the manuscript, there are inconsistencies when referring to the EMCs. Since the EMC acronym is introduced, I would recommend replacing all instances of "cyclone" with "EMC." Additionally, saying EMC "member" seems unnecessary.

L134: Stating that "six clusters were found" is incorrect. The 6 SOM nodes were chosen by the user. In the context of SOM studies, the word "cluster" is commonly used when different SOM "nodes" are grouped together to help describe the scientific results. Alternatively, you could say: "A 2 X 3 SOM (6 nodes) was used for this study." Additionally, more explanation as to why only 6 nodes were chosen as well as why a rectangular and non-square SOM was used. What type of topology was used, what neighborhood function, what distance metric?

L135: Were there any subtle differences when utilizing more SOM nodes? While the PV maps may not look very different, are there notable differences in precipitation and temperature anomalies? Using a SOM is a great way to pick out subtle differences, but that can be difficult if you have too few nodes.

L139-149: While it is good that you ran the SOM algorithm on the data multiple times, it would also be useful to discuss other metrics (that are not subjective) to show that the 2 by 3 SOM is a good choice for this data, such as the quantization error the correlation of the different maps to their categorized SOM node.

**2.5 CYCLONE DEEPENING RATE**

L152-153: Can you quantify short vs long tracks?

**2.6 SIGNIFICANCE OF THE TRENDS**

L165: Is the rainy season DJF? A clarification in parenthesis would be beneficial. Also, I want to make sure I understand this correctly, the 1979 and 2020 EMCs are included, but are they just not used for the MK trend analysis?

**3 RESULTS**

**3.1 CYCLONE CHARACTERISTICS**

L169: Is there a duration requirement for how long the EMCs persist within the smaller study domain?

**3.2 SOM CLASSIFICATION**

L177: SOM groupings are not "revealed" they were "chosen" by the user

L178: You can talk about how the neighboring nodes share similar PV distributions because the SOM is useful for spatial data and showing spatial relationships. Again, say "nodes" instead of clusters.

Entire manuscript: label SOM nodes with [row #, column #] instead of saying "cluster #." This is consistent with the SOM literature and makes it easier for the reader to visualize the SOM as it is being described in the text.

L182: Add "a" between "and" and "less"

L183: Replace "members" with "EMCs" and only have "22%" in the parentheses. Ensure this is consistent throughout the manuscript.

L186: For node 6, there is still yellow PV shading which indicates that the PV values are still of the same magnitude as the other nodes. Consider revising the sentence with more specific geographic references.

L193-194: Have you already created EMC-centered composites? I think that would help clean up a lot of the noise that can be seen in the figures.

**3.3 SEASONALITY**

Fig. 3: I found it fascinating that each SOM node features a different month with its highest frequency of occurrence. To improve communication of these results, you could create histograms like Fig. 1c for each SOM node. That way, the reader can connect back to the spatial relationships between SOM nodes.

L209: Add "...as winter progresses into spring," especially because node 4 EMCs occur the most frequently in April.

L216-236: I liked how you added the numerical values that describe the EMC distance when referenced throughout the text (for example, L219). I think it would also help the reader if you did that for the deepening rates and duration.

L220: Adding a figure that shows EMC tracks for each SOM node would further support this statement as well as complement Table 1.

L229: This paragraph could go with the previous, unless a transitionary sentence is added to the start of it.

L230: "Sharav" does not need to be in quotes.

Table 1: A mention of these values being the average values for the EMCs in each node would help clarify this table and caption.

L244: Well done mentioning that this is important for assessing surface impacts.

L247: I like the specific geographic features and regions mention here, please do this more throughout the manuscript (instead of saying domain).

Other thoughts: A sentence discussing how the IMERG data compares in regions with convection dominated ERA5 precipitation would be a good benefit to the paragraph starting on L256.

L266: Remove "significant" and listing the types of precipitation hazards with strengthen this paragraph.

L288: "The fact that" and "interesting" can be removed from this sentence to be more concise.

L294-295: This is a great connection, and you can also mention that this can also be seen in Table 1.

L301: Refer back to Fig.2

Other thoughts: A figure that shows mid-level ascent/descent as well as moisture availability would further support the connection between the PV SOM nodes and the observed precipitation.

**3.4 PRECIPITATION DISTRIBUTION, VARIABILITY, AND EXTREMES**

L319: Since moisture availability is not shown for the nodes, I am not sure if saying that a node "drier" is the best way to describe a node with less precipitation.

L324: abrupt change to talking about localized precipitation in Israel. Maybe start a new subsection?

Figure 7: As someone not as familiar with the geography of Israel, could you add more geographic information (cities, topographic contours, etc.) to orient themselves?

L332: Stating that "Cluster 5 extremes dominate southern Israel" is too generalized given that there is only one observation site there.

Table 2: I am struggling with understanding this table. What are rank-1, rank-2, and rank-3? Why don't all of the values in a column add up to 60? The explanation in the caption would benefit from improved clarity.

L368-373: When reading the previous paragraph, I had questions about if showing the +12 hr plots were just showing diurnal patterns, so I am glad you addressed the consistency of these patterns in this paragraph!

**3.6 OBSERVED CYCLONE FREQUENCY TRENDS**

Note: This section may be best located after the initial SOM is introduced.

L380: Can you also include the m-values in Fig. 1a like you did in Fig. 9 to support this statement?

Figure 9: Awesome figure!

L384: I really like this sentence.

L386-388: Yes!

Other note: Why do you think there is such an oscillation pattern for Cluster 5? It is almost interdecadal. Maybe ENSO? Curious about what you think of this.

**4 DISCUSSION AND CONCLUSIONS**

Figure 10: What do the different colors on the upper panel represent?

L391: "grouped"/ "categorized" may fit better than "classified"

L393: More discussion about temperature anomalies would benefit this list. I like how you also talked about significant trends in Cluster 5, I think it would be great to do that for each node.

L424: If the goal of this manuscript is to provide generalized information about EMCs to forecasters, then I actually think that EMC-centered composites would be more beneficial. Centered composites would provide clearer signals that forecasters can think about and use to improve their forecasts.

L430-432: I disagree. There needs to be a discussion about the subjectivity of choosing how many SOM nodes and the structure of the SOM.

Other note: The localized precipitation in Israel was not discussed in this concluding section—is this supposed to be a key part of the manuscript?